# Proteome Analysis of Alpine Merino Sheep Skin Reveals New Insights into the Mechanisms Involved in Regulating Wool Fiber Diameter

**DOI:** 10.3390/ijms242015227

**Published:** 2023-10-16

**Authors:** Lin Yue, Zengkui Lu, Tingting Guo, Jianbin Liu, Bohui Yang, Chao Yuan

**Affiliations:** 1Lanzhou Institute of Husbandry and Pharmaceutical Sciences, Chinese Academy of Agricultural Sciences, Lanzhou 730050, China; 2Sheep Breeding Engineering Technology Research Center of Chinese Academy of Agricultural Sciences, Lanzhou 730050, China; 3Key Laboratory of Animal Genetics and Breeding on Tibetan Plateau, Ministry of Agriculture and Rural Affairs, Lanzhou 730050, China

**Keywords:** Alpine Merino sheep, proteomics, wool fiber diameter, WGCNA

## Abstract

Wool fiber is a textile material that is highly valued based on its diameter, which is crucial in determining its economic value. To analyze the molecular mechanisms regulating wool fiber diameter, we used a Data-independent acquisition-based quantitative proteomics approach to analyze the skin proteome of Alpine Merino sheep with four fiber diameter ranges. From three contrasts of defined groups, we identified 275, 229, and 190 differentially expressed proteins (DEPs). Further analysis using Gene Ontology (GO) and Kyoto Encyclopedia of Genes and Genomes (KEGG) pathways revealed that pathways associated with cyclic adenosine monophosphate and peroxisome proliferator-activated receptor signaling are relevant to wool fiber diameter. Using the K-means method, we investigated the DEP expression patterns across wool diameter ranges. Using weighted gene co-expression network analysis, we identified seven key proteins (CIDEA, CRYM, MLX, TPST2, GPD1, GOPC, and CAMK2G) that may be involved in regulating wool fiber diameter. Our findings provide a theoretical foundation for identifying DEPs and pathways associated with wool fiber diameter in Alpine Merino sheep to enable a better understanding of the molecular mechanisms underlying the genetic regulation of wool fiber quality.

## 1. Introduction

Alpine Merino sheep are a high-altitude, cold, and drought-resistant fine-wool breed cultivated in China. The wool fiber diameter of Alpine Merino sheep is typically 19–21.5 μm, and the breed exhibits great economic potential among plateau animals. As a natural, renewable, and biodegradable high-quality fiber [1], wool, especially superfine wool, is favored in the textile industry. Therefore, studying the biological regulatory mechanisms of wool fiber diameter is important for improving the economic value of wool and promoting the development of the fine-wool sheep industry.

Research on wool fiber diameter is generally conducted from two aspects. One aspect involves exploring the relationship between wool fiber diameter and a single gene or locus, then determining it as a molecular marker for subsequent breeding programs. The second aspect involves simultaneously exploring multiple genes that may affect wool fiber diameter. Omics research is more in line with the characteristics of micro-effect polygenes corresponding to quantitative traits such as wool fiber diameter. Studies on a single gene or locus mainly focus on the two protein families of keratin (KAP) and keratin-12-associated proteins (KRTAP; e.g., KRTAP1-3, KRTAP2-1, KAP24-1, and KAP27-1) [2,3,4,5]. Genes such as *EEF1D* and *MYL6* have also been associated with wool fiber diameter [6,7]. Few reports have been published on the omics of wool fiber diameter, and many molecular regulators are involved in hair follicle development. Wang et al. [8] studied the DNA methylation group and transcriptome of Cashmere goat skin to reveal the molecular regulatory mechanism of hair follicle morphogenesis and skin development. Zhao et al. [9] used the whole transcriptome and DNA methylation group to determine the signaling pathways, key genes, and transcription factors of developmental stages of hair follicles. He et al. [10] used RNA-seq technology to screen genes related to hair follicle development in Subo Merino sheep and analyzed the expression regulation patterns of these genes. Zhao et al. [11,12] directly explored the regulatory mechanism of Cashmere fineness in Tibetan Cashmere goats through omics and combined the transcriptome, proteome, IncRNAs, and mRNA to analyze the genetic mechanisms of Cashmere traits and key candidate biomarkers.

Here, wool fibre diameter was divided into four different grades and three control groups were set up according to the gradient of wool fibre diameter, with the intention of searching for the differences in proteins exhibited in the different gradient ranges, and using the differences in proteins in the three control groups to search for the biological mechanisms regulating the diameter of wool fibres of different gradients, in order to provide theoretical references for the gradual optimisation of wool fibre diameter. Our findings enable a better understanding of the genetic background of wool fiber diameter differences and the mechanisms involved in regulating wool fiber diameter. Our results provide a reference for the development of fine-wool sheep breeding and the fine-wool industry.

## 2. Results

### 2.1. Differential Protein Analysis

According to the difference of wool fiber diameter (FD), coefficient of variation (CV), and mean fiber diameter (MFD), we divided the experimental individuals into four gradients, SF group (FD < 18.0 μm, CV < 20%, MDF = 17.68 ± 0.26 μm), EF group (18.0 μm ≤ MFD < 20.0 μm, CV < 22%, MFD = 19.15 ± 0.37 μm), MF group (20.0 μm ≤ MFD < 21.5 μm, CV < 22.7%, MDF = 20.67 ± 0.31 μm), and M group (21.5 μm ≤ MFD < 23.0 μm, CV < 23.6%, MDF = 22.51 ± 0.43 μm), and set up three control groups EF/SF, MF/EF, M//MF. We identified 9318 plausible proteins from these three control groups (Appendix A) and analyzed the expression levels of these proteins (Appendix A). The EF/SF group contained 275 DEPs, of which, 143 were upregulated and 132 were downregulated (Figure 1A). The MF/EF group contained 229 DEPs, of which, 150 were upregulated and 79 were downregulated (Figure 1B). The M/MF group contained 190 DEPs, of which, 110 were upregulated and 80 were downregulated (Figure 1C). The Venn diagram (Figure 1D) shows 64 common proteins between the EF/SF and MF/EF groups, 29 common proteins between the MF/EF and M/MF groups, and 4 common proteins among all three groups. Counting the expressions of the four common proteins revealed that the expressions of LOC114109328 and EML2 in each group were significantly higher than those of BCL7C and PAPPA (Figure 1E).

### 2.2. Functional Analyses of Identified DEPs

We performed GO and KEGG functional analyses of all DEPs in the three comparison groups. GO enrichment analysis showed that the DEPs in the EF/SF group in the biological process category were mainly involved in lipid metabolism, glycerol-3-phosphate catabolism, and extracellular matrix tissue. DEPs in the MF/EF group were mainly involved in lipid metabolism and enzyme activity regulation. DEPs in the M/MF group were mainly involved in signal transduction, followed by transmembrane transport. In the cell components, all three groups were enriched in keratin filaments, and the EF/SF group was enriched in endoplasmic reticular membrane and intermediate filaments. The MF/EF group was enriched in extracellular space and transcription factor tfIIa complex. The M/MF group was enriched in transport particles A and B in the ciliary body. DEPs in the molecular functional category were mainly related to molecular binding and molecular activity, including nucleic acid binding, NAD binding, and structural molecular activity in the EF/SF group, and lipid binding, oxygen binding, and oxygen carrier activity in the MF/EF group. Transcription factor binding and glutathione peroxidase activity were enriched in the M/MF group (Figure 2, Appendix A). KEGG enrichment analysis showed that most EF/SF groups were enriched in metabolic pathways, including glycerophospholipid metabolism, fat metabolism, amino acid metabolism, regulation of lipolysis of adipocytes, fat digestion, and absorption. Abundant pathways in the MF/EF group included metabolic pathways, RNA polymerase, the cyclic adenosine monophosphate signaling pathway, and the ECM–receptor interaction and pathway. Peroxisome proliferator-activated receptor (PPAR) signaling pathway, nuclear transport, protein digestion and absorption, and cholesterol metabolism were the main enriched pathways in the M/MF group (Figure 3, Appendix A).

We evaluated the GO terms related to wool fiber diameter and DEPs in the KEGG pathways. GO terms included keratin filaments, intermediate fibers, and structural molecular activity. Related proteins included A0A836AN55, K38, KRT28, KRT39, A0A6P3E7M5, A0A835ZRC9, W5QIW0 (Figure 4A). KEGG pathway analysis revealed that A0A836D5A1, A0A835ZPD9, LAMA3, CAMK2G, LAMA2, PPP2R1A, and CHD8 were enriched in the PI3 kinase-Akt signaling pathway, Wnt signaling pathway, and TGF-beta signaling pathway (Figure 4B). These pathways are closely related to hair follicle development and indirectly affect the wool fiber diameter.

### 2.3. K-Means Analysis of DEPs

To find the DEP expression patterns among wool fiber diameter ranges, we performed K-means and GO enrichment analyses. DEPs in each cluster had different expression patterns and were enriched to different GO terms (Figure 5, Appendix A). DEPs in cluster 1 were significantly enriched in the regulation of ARF protein signal transduction and maintenance of protein localization in the nucleus. Protein expression in cluster 1 was highest in the MF group, and expression levels did not differ significantly among the other three groups. The DEPs in cluster 2 were mainly enriched in calcium ion binding, cell attachment and mannooligosaccharide α-1,2-mannosidase activity, and protein expression in this cluster was lowest in the EF group. DEP expression in cluster 3 was lowest in the SF group, but its enriched terms were relatively rich and mainly included transferase activity, structural molecular activity, intermediate filaments, and cell cold response. DEPs in cluster 4 were enriched in oxygen transporter activity and coagulation regulation, and the expression level was the lowest in the MF group.

### 2.4. Weighted Gene Co-Expression Network Analysis

We used standardized data from 517 DEPs to construct an expression matrix and then divided it into four modules (Figure 6). The turquoise module contained the most DEPs, but the correlation between this module and fiber diameter was low. For the MF group, protein expression was significantly higher in the yellow module than in the other modules, but the expression levels in the SF, EF, and M groups were very low, and the degree of correlation with fiber diameter was low. Protein expression in the blue module was relatively low in each group, and the degree of correlation with fiber diameter was low. Proteins in the brown module were highly correlated with fiber diameter; thus, we performed GO analysis on the highly expressed proteins in the yellow module and the highly correlated proteins in the brown module. Enriched proteins in the brown module were relatively rich, those in the yellow module were not (Appendix A). In the BP category, proteins in the brown module were mainly involved in the positive regulation of cell response, organ growth, and immunoglobulin production, and proteins in the yellow module were mainly involved in the regulation of ARF protein signal transduction. In the CC category, proteins in the brown module were mainly involved in the composition of endoplasmic reticulum membrane and fat granules. Proteins in the MF category were mainly involved in transferase activity and protein homodimerization activity. Proteins in the yellow module were mainly enriched in ribosome binding. We also analyzed the top 25 proteins with the highest connectivity in the brown and yellow modules and show the relationship between them below (Figure 7).

## 3. Discussion

Wool fiber diameter is an important economic trait in sheep, and studying its molecular regulatory mechanisms is the most effective approach to optimizing wool fiber diameter. Proteins often directly reflect biological functions, therefore, studying proteomics enables a better understanding of the potential molecular mechanisms involved in regulating wool fiber diameter. We found four common proteins among three comparison groups. Of these proteins, LOC114109328 and EML2 were highly expressed in each group. EML2 plays an important role in mitosis, mainly in forming the spindle and interphase microtubule network, and often appears in cancerous tumor-related research as a tumor suppressor [13,14,15]. Both BCL7C and EML2 have similar functions as tumor suppressors, and both are related to islet β cells [16]. PAPPA differs from BCL7C and EML2; its overexpression promotes tumor growth [17], and it is related to IGFBP hydrolysis and activation of NF-κB, PI3K, and other pathways [18], which are closely related to hair follicles and wool fibers. Therefore, we speculate that PAPPA is likely involved in regulating wool fiber diameter. Combined with PAPPA’s expression in each group, the difference in fiber diameter between the EF and MF groups may be related to PAPPA’s expression difference in these groups, but the specific regulatory mechanism requires further exploration.

We identified 694 DEPs, a considerable number compared with those of previous reports [11]. The GO terms of these DEPs were rich and included biomacromolecular binding, carrier activity, and signal transduction. Among them, keratin filaments and intermediate filaments were significantly enriched in these groups. The intermediate filament is composed of keratin, mainly as a cytoskeleton, and exists in all cell types [19,20]. The intermediate filament in the epithelium is called the keratin filament, which is the main component of the cortex and the basis of wool fiber structure [21,22]. Cortical tissue is closely related to wool fiber fineness. Studies have shown that as the fiber diameter increases, the proportion of orthocortex in wool fibers increases, while the proportion of paracortex decreases [23]. This further demonstrates the important role of keratin filaments in determining wool fiber diameter. Keratinocyte proliferation is also important. It involves many regulatory factors, such as growth factors, fat-soluble vitamins and their derivatives, neuropeptides, neurotransmitters, and steroids, which may be involved in regulating keratinocyte growth [24,25]. Some substances have a specific effect on wool fiber diameter [26,27,28]. KEGG analysis showed that the PPAR signaling pathway was significantly enriched in the M/MF group. PPAR is a member of the nuclear hormone receptor family and plays an important role in lipid metabolism in sebaceous glands [29,30]. Evidence also suggests that sebaceous glands and sweat glands affect hair follicle morphogenesis, thus linking the PPAR pathway to hair follicle development [31,32,33]. PPAR is also involved in regulating keratinocyte differentiation and forming functional skin barriers [34,35]. Studies have shown a potential link between the PPAR signaling pathway and wool fibers. We compared the KEGG pathways enriched in each group and found that most of the DEPs in the EF/SF group were enriched for amino acid metabolism, and that amino acids are mostly interpreted from a nutritional point of view during hair growth. Therefore, we hypothesized that the difference in wool fibre diameter between the EF/SF groups could be related to nutritional status. There are several pathways affect-ing hair follicle development in the MF/EF and M/MF groups. Reducing the differences in wool fiber diameters between the MF/EF and M/MF groups is currently a key issue. Our results may help resolve this.

K-means analysis showed that the DEPs in each cluster performed differently in different fiber diameter ranges. Specifically, MF histone expression was significantly higher in cluster 1 than in the other groups. Functional enrichment analysis showed that the DEPs in cluster 1 were significantly enriched in the regulation of ARF protein signal transduction and maintenance of protein localization in the nucleus. ARF protein signal transduction occurs in nearly every cellular biological process and interacts either synergistically or antagonistically with TGF-β, MAPK, WNT, and other signaling pathways [36,37,38,39]. Studies have described the relationship between these classical pathways and hair follicle development [40,41,42]. We speculate that ARF protein signal transduction may also be involved in hair follicle development. The MF histone expression in cluster 1 and the difference in wool fiber diameter may be related to this pathway. DEP expression patterns were similar in the other three clusters, and all were enriched in GO terms directly or indirectly related to wool fiber. However, in addition to these terms, terms such as coagulation regulation and cell cold response were also enriched, which may indicate the adaptability of Alpine Merino sheep to the cold and arid regions of the plateau.

The brown module had the highest degree of correlation with wool fiber diameter. We searched for key proteins that may regulate wool fiber diameter in this module. Enrichment analysis revealed that protein homodimerization activity is closely related to wool fibers. As mentioned, keratin constitutes wool fibers. Its structure is closely related to protein homodimerization activity. Intermediate filaments are also formed based on the α-helix domain in keratin [43,44,45], and its structural changes may affect wool fiber diameter. Therefore, we focused on the proteins including CIDEA, CRYM, MLX, TPST2, GPD1, GOPC, and CAMK2G. Previous studies suggest that these proteins are mainly involved in metabolic and signal transduction processes. CIDEA [46], CRYM [47], and MLX [48] regulate fat metabolism through different mechanisms, and GOPC [49] and CAMK2G [50] affect disease occurrence by participating in cell signal transduction. Currently, these proteins show a direct relationship to wool fiber diameter. Our results differ from those of previous studies [11], however, some of our proposed proteins have not previously been mentioned and further research is warranted. Furthermore, we discovered that the DEPs in these two modules were enriched in items linked to cellular response to cold and the regulation of ARF protein signal transduction. This finding strongly suggests that the regulation of ARF protein signal transduction may have a significant impact on wool fibre diameter regulation. The cell’s response to cold stems from DEPs enrichment in the blue module. It suggests that this response may explain the adaptability of alpine Merino sheep to high-altitude cold and drought areas, as well as possibly being connected to the diameter of wool fibers. This closely aligns with the findings of Zhao et al. [11], who investigated the correlation between wool fibre breadth and animal stress and hypoxia in alpine Merino sheep living in high-altitude conditions. In addition to the stress factor of hypoxia, cold is al-so a significant stress factor. As such, the cellular response to cold is most likely implicated in the regulation of wool fibre diameter.

In conclusion, our study of the proteomics of Alpine Merino sheep skin has revealed several significant signaling pathways, although further validation is necessary for specific pathways. It’s important to recognize that the regulation of wool fiber diameter is a complex and multifaceted process that warrants more diverse approaches than simply focusing on already discovered classical pathways. Further research is needed to improve the understanding of wool fibre diameter regulation processes and to provide practical methods for their optimization.

## 4. Materials and Methods

### 4.1. Weighted Gene Co-Expression Network Analysis

The Alpine Merino sheep used in this experiment originated from the Gansu Sheep Breeding and Promotion Station (Huangcheng Town, Zhangye City, China). All of the experimental animals came from the same population. According to the difference of wool fiber diameter (FD), coefficient of variation (CV), and mean fiber diameter (MFD), we divided them into SF group (FD < 18.0 μm, CV < 20%, MDF = 17.68 ± 0.26 μm), EF group (18.0 μm ≤ MFD < 20.0 μm, CV < 22%, MFD = 19.15 ± 0.37 μm), MF group (20.0 μm ≤ MFD < 21.5 μm, CV < 22.7%, MDF = 20.67 ± 0.31 μm), and M group (21.5 μm ≤ MFD < 23.0 μm, CV < 23.6%, MDF = 22.51 ± 0.43 μm). Six sheep were collected from each group, and the skin tissue was derived from the shoulder of the experimental animal. The collection area was about 2 cm × 3 cm, was placed in a freezing tube, immediately frozen in liquid nitrogen, and then stored at −80 °C. All animal experiments were performed under the guidance of ethical regulations from the Institutional Animal Care and Use Committee of Lanzhou Institute of Husbandry and Pharmaceutical Science of the Chinese Academy of Agricultural Sciences (Approval No. NKMYD201805; Approval Date: 18 October 2018).

### 4.2. Skin Tissue Protein Extraction and Digestion

The skin tissues of Alpine Merino sheep were fully ground to powder in liquid nitrogen, and the sample lysate (Biyuntian), phosphatase inhibitor (Roche, Basel, Switzerland), and protease inhibitor PMSF (Biyuntian) were added to make the final concentration of the solution 1 mM. The cold research instrument (Shanghai Wanbai Biotechnology Co., Ltd. Shanghai, China) was ground at −35 °C, 60 Hz, 120 s and repeated once. The solution was centrifuged at 12,000 rpm for 10 min at 4 °C, and the supernatant was taken. The supernatant was the total protein solution of the sample. The protein concentration and molecular weight were determined by the BCA method and SDS-PAGE method, respectively, and stored at −80 °C for later use. According to the measured protein concentration, an appropriate amount of the sample was taken and diluted to the same concentration and volume. Thereafter, 5 mM DTT was added and incubated at 55 °C for 30 min, then cooled to room temperature. We then added 10 mM iodoacetamide and left it in a dark at room temperature for 15 min. Following that, we added 6 times the volume of acetone to precipitate the protein and placed it at −20 °C for 4 h. We then centrifuged to collect precipitate and volatilize acetone. The precipitate was redissolved with 50 mM NH_4_HCO_3_, and 1 mg/mL trypsin of 1/50 sample mass was added to digest at 37 °C overnight. The pH value was adjusted to about 3 with phosphoric acid in order to terminate the enzymatic hydrolysis reaction.

### 4.3. High-Performance Liquid Chromatography

The separation was performed on an 1100 HPLC System (Agilent, Palo Alto, CA, USA) using a Nano Chrom-C18 column (5 μm, 150 mm × 2.1 mm). Mobile phases A (2% acetonitrile in HPLC water) and B (90% acetonitrile in HPLC water) were used for gradient. The solvent gradient is shown in Appendix A. Tryptic peptides were separated at a fluent flow rate of 250 μL/min and monitored at 210 nm. Samples were collected for 10–50 min, and the eluent was collected in centrifugal tubes every 1–10 min in turn. The samples were then recycled in this order until the end of the gradient. The separated peptides were lyophilized for mass spectrometry.

### 4.4. Mass Spectrometry Analysis

Nanoflow reversed-phase chromatography was performed on an EASY-nLC 1200 system (Thermo Fisher Scientific, Waltham, MA, USA). Peptides were separated at 90 min at a flow rate of 300 nL/min on a 25 cm × 75 μm column (1.6 μm C18, ionopticks). Mobile phases A and B were 0.1 vol% formic acid solution and 80:20:0.1 vol% ACN: water: formic acid, respectively. The total run was 60 min (045 min, 5–27% B; 45~50 min, 27–46% B; 50~55 min, 46–100% B; 55~60 min, 100% B). Liquid chromatography was coupled online to a hybrid TIMS quadrupole TOF mass spectrometer (Bruker timsTOF Pro, Saarbrücken, Germany) via a CaptiveSpray nano-electrospray ion source. The conditions of DDA mass spectrometry are shown in Appendix A. To perform Data-independent acquisition (DIA), we used the instrument control software (Bruker otofControl v6, Saarbrücken, Germany) to define quadrupole isolation windows as a function of the TIMS scan time (diaPASEF). The conditions of DIA mass spectrometry are shown in Appendix A.

### 4.5. Proteome Data Processing Analysis

Spectronaut (Version 15.3.210906.50606) was used to search all of the raw data thoroughly against the sample protein database. Database search was performed with Trypsin digestion specificity. Alkylation on cysteine was considered a fixed modification in the database search. Protein, peptide, and PSM’s false discovery rate (FDR) were all set to 0.01. For DIA data, the quantification FDR was also set to 0.05. Quantity MS-level was set at MS2. The proteins that had a fold change of 1.5 and a *p*-value < 0.05 in three comparison groups (EF/SF, MF/EF, M/MF) were considered DEPs.

We performed gene ontology GO and KEGG analysis on DEPS in these three groups. Go functional annotation was used http://www.geneontology.org (accessed on 9 August 2023), KEGG functional annotation was used https://www.kegg.jp/ (accessed on 10 August 2023), GO and KEGG pathways *p* < 0.05). In the STRING database, the species/related species (blast e-value: 1 × 10^−10^) were selected to analyze the differential proteins, and the interaction relationship of the differential proteins was obtained.

### 4.6. K-Means Analysis and WGCNA

The K-means function in the R package (version 4.2.1) was used to cluster all DEPs, where k = 4. We interpolated the missing values of proteome data based on the predictive mean matching (PMM) method in multiple interpolations of chain equations (MICE) [51] and filtered the proteins with a low fluctuation of expression level (standard deviation ≤ 0.5). Then, WGCNA analysis was performed and the power value was set to 10. Then, the weighted gene co-expression network was constructed and divided into modules. According to the absolute value of the correlation coefficient greater than or equal to 0.3 and the *p*-value less than 0.05 as the threshold, the modules related to traits were screened. The core proteins in the modules closely related to the diameter of wool fiber were identified and the protein interaction network was drawn.

## 5. Conclusions

We analyzed the skin proteome of Alpine Merino sheep and identified previously unreported signaling pathways. We used multiple analytical methods to identify proteins that may be involved in regulating wool fiber diameter. These results further enhance our knowledge of the mechanisms underlying wool fiber diameter regulation and provide a reference for future studies in analyzing these mechanisms. This work provides a theoretical foundation for breeding fine-wool sheep.

## Figures and Tables

**Figure 1 ijms-24-15227-f001:**
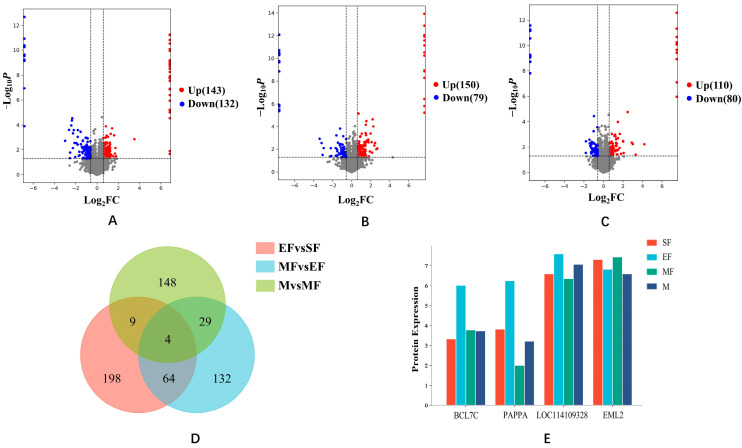
Differential protein expression in the three comparison groups. In the (**A**–**C**) volcano plots, red dots corresponded to up-regulated proteins (FC ≥ 1.5, *p* < 0.05), and blue dots corresponded to down-regulated proteins (FC ≤ 0.66, *p* < 0.05), Grey dots correspond to insignificant proteins. ((**A**): EF/SF, (**B**): MF/EF, (**C**): M/MF). (**D**) The Venn diagram represents the DEPs comparison between the three groups. (**E**) The expression of the four common proteins in each group.

**Figure 2 ijms-24-15227-f002:**
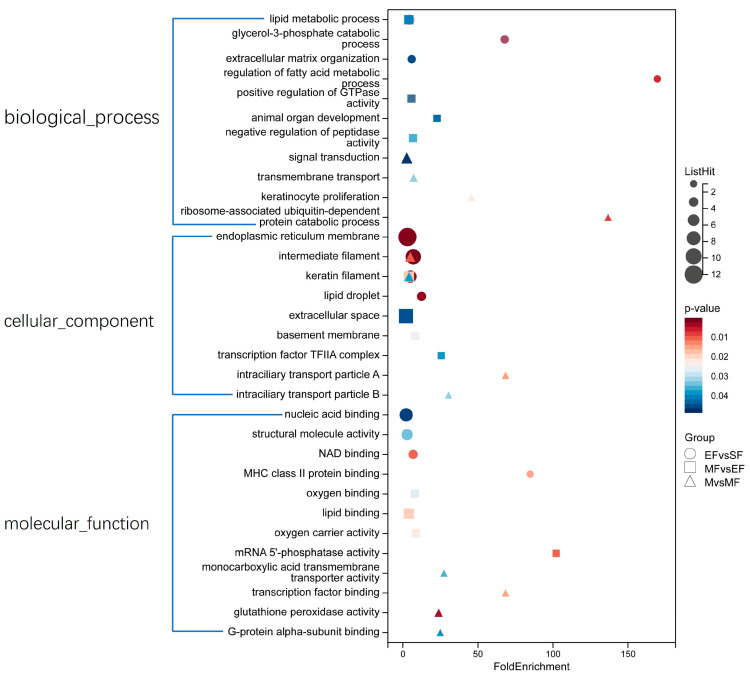
GO enrichment analyses of DEPs.

**Figure 3 ijms-24-15227-f003:**
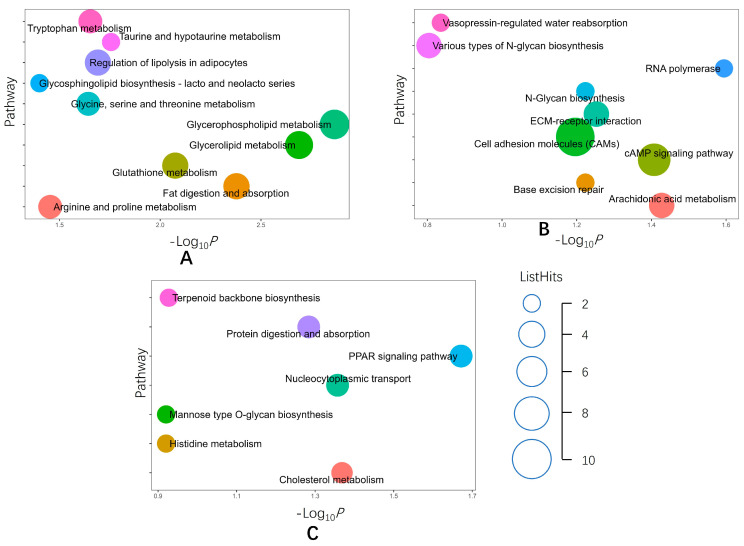
KEGG enrichment analyses of DEPs. (**A**) KEGG enrichment analysis of differentially expressed proteins in the EF/SF group. (**B**) KEGG enrichment analysis of differentially expressed proteins in the MF/EF group. (**C**) KEGG enrichment analysis of differentially expressed proteins in the M/MF group.

**Figure 4 ijms-24-15227-f004:**
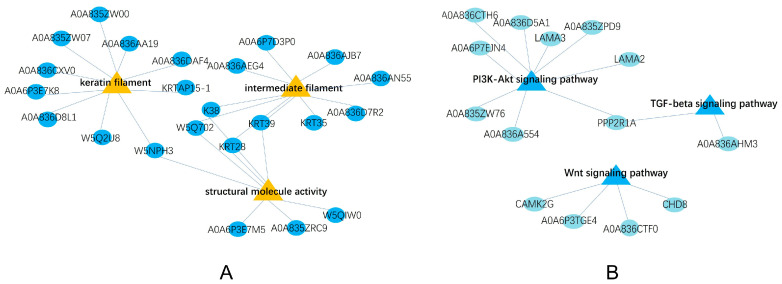
(**A**) PPI network of wool fiber diameter-related GO terms (yellow). (**B**) PPI network of wool fiber diameter-related KEGG pathways (blue).

**Figure 5 ijms-24-15227-f005:**
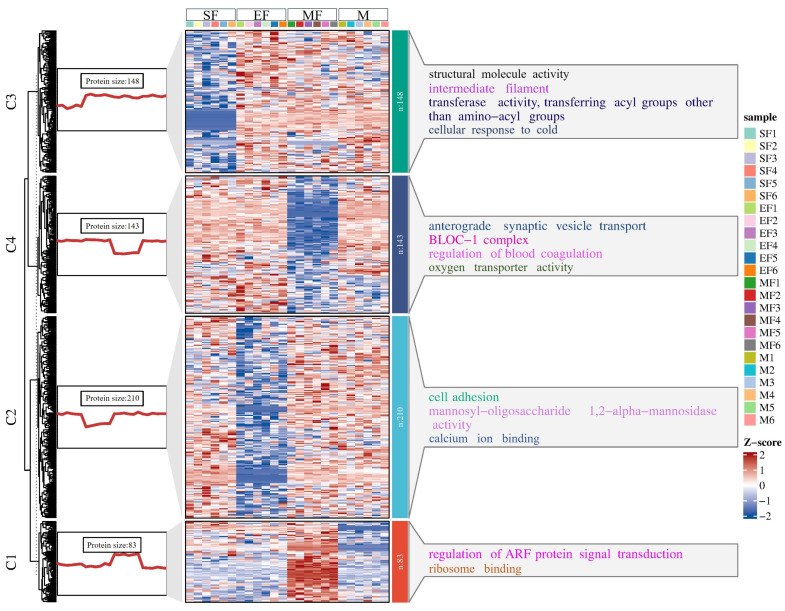
Expression trend map and heat map. The left side is the overall expression trend of differential proteins in each cluster after K-means analysis, the middle is the expression of differential proteins in each cluster, and the right side is the GO terms enriched by differential proteins in each cluster.

**Figure 6 ijms-24-15227-f006:**
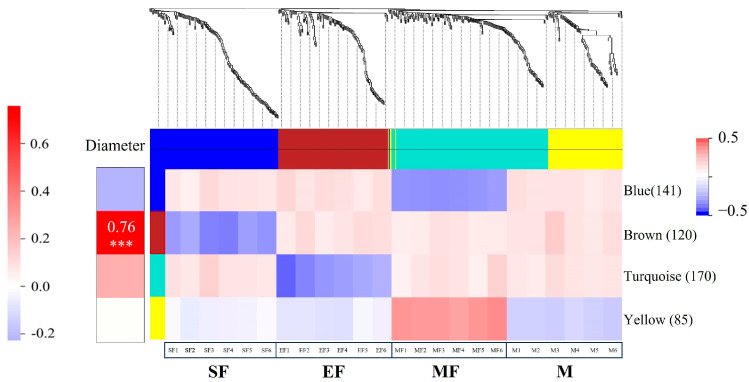
Module division and its correlation with samples, DEP is divided into four modules: blue module, brown module, blue–green module, and yellow module. The left side is the result of correlation analysis between each module and fiber diameter, The deeper the color, the stronger the correlation. *** indicates strong correlation.

**Figure 7 ijms-24-15227-f007:**
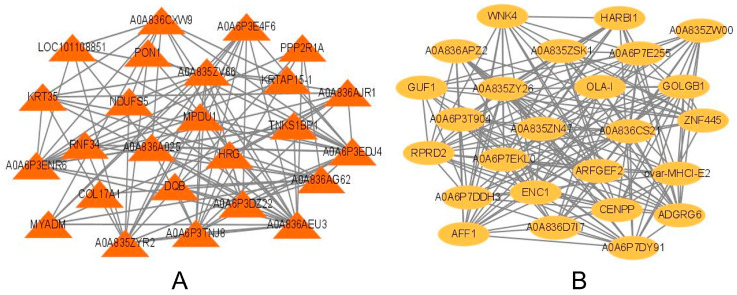
Core protein network diagram. (**A**) is a brown module, and (**B**) is a yellow module.

## Data Availability

The mass spectrometry proteomics data have been deposited to the ProteomeXchange Consortium http://proteomecentral.proteomexchange.org (accessed on 9 October 2023) via the iProX partner repository with the dataset identifier PXD046045.

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
