# Peer review of "Proteome Analysis of Alpine Merino Sheep Skin Reveals New Insights into the Mechanisms Involved in Regulating Wool Fiber Diameter"

_ijms, 2023, doi:10.3390/ijms242015227_

Round 1

Reviewer 1 Report

Interesting study, but I would like some more detailed exaplenations.

Firstly authors refear to different groupd EF, SF, MF - it was confusing to have refeared to this before the explanation of what it is - eventually i found it in methods, but it wasnt obvious.

2. How many proteins were detected for each of these groups. I am suprirsed that there is such a low overlap between different groups represented here. Why is this the case if its the same spiecies and same sample processing. only the quantification of the proteins should change and not the identities.

3. Proteomics data should be deposited in the PRIDE repository before manuscript is published.

4. I would like a bit more discussion in terms of the recent publications in the field. A quick scholar search revealed another proteomics of wool fibers study that is not discussed in context of this manuscript:The wool proteome and fibre characteristics of three distinct genetic ovine breeds from Portugal 

5. it wasnt obvious reading the manuscript initially how big are the sample sizes in each of the groups. this should be highlighted.

6. What database search was used? Authors do mention they used Spectronaut - but is this alligned against a specific uniprot database -only reviewed or anso unreviewed protein entries were used?

7. I would also like to see Padog Reactome analysis performed on this study.

Didnt see any issues, everything was well writted. some typos in some places may need to be adresssed.

Reviewer 2 Report

Please review and make adjustments as necessary to reflect this study investigated proteins and did not evaluate gene expression. There are numerous instances of DEG when it should be DEP, including but limited to figure captions, subsection headings, etc. given you do not have gene expression data reported.

L17: From three comparisons, not three control groups.

L46: Genes should italicized, EEF1D and MYL6

L68: The presentation of your results could be clearer if you refer to the comparison as comparisons and not as groups. It makes it harder for the reader to follow which group(s) you are referring to since you also define four groups of fiber characteristics.

L79: Fig 1. Please adjust the axis’s of panels A-C to properly show the placement of DEPs which are currently congregated at the vertical edges of each plot. I suspect those DEPs do not all have a value of -8 or 8 fold change. I recommend adding to the caption for panel E, “the expression of the four common proteins in each group” to be consistent in your language from the text and clarify why you are presenting these four proteins.

L124: Figure 4: Do the colors represent anything, if so please provide a caption, if not I’d make the colors consistent between panels A & B.

L145: Section 2.4: I recommend renaming your clusters or label figure 6 appropriately for those that may be color-blind.

L170: gene, should be protein in Figure 7 caption.

L235: I’m assuming on should be no

L240: Were the animals in each group related to one another or selected to be less related?  

L243: Please describe why you chose the cutoffs you did for your sampling groups? As well as place in the text what SF, EF, MF, and M are meant to represent.

L246: The scapula is the name of the bone. Please rephrase to make clearer that the skin sample was collected from the shoulder area.  

L307: k-means sub-section heading spelling

L308: specify and cite which R package was used.

Some minor spelling and grammatical errors to change throughout.

Round 2

Reviewer 2 Report

Thank you for the revisions

Some minor grammatical issues can be caught in proofing

Author Response

Thank you very much for your suggestions on this manuscript, we have proofread the content of the manuscript and have revised it for grammatical problems, thank you again for reviewing this manuscript.
